## Research Article

coastal storm; backshore erosion; storm impacts; beach erosion; foredune erosion

**Corresponding author:**
Thomas S. N. Oliver;
Email: t.oliver@unsw.edu.au

# Foredune erosion, overtopping and destruction in 2022 at Bengello Beach, southeastern Australia

Thomas S. N. Oliver[1] 📧, Michael A. Kinsela[2], Thomas B. Doyle[3,4] 📧 and Roger F. McLean[1]

[1]University of New South Wales at ADFA, Canberra, ACT, Australia; [2]School of Environmental and Life Sciences, University of Newcastle, Callaghan, NSW, Australia; [3]Department of Climate Change, Energy, the Environment and Water, Sydney, NSW, Australia and [4]School of Earth, Atmospheric and Life Sciences, University of Wollongong, Wollongong, NSW, Australia

## Abstract

The beach–foredune system at Bengello Beach has been monitored monthly to bimonthly at four profiles (P1–P4) since 1972 and documented the building of a foredune. This paper addresses the remarkable changes which occurred in 2022 as storm waves overtopped and trimmed this foredune at all profiles, then later removed this entire feature at two of the profiles (P3, P4) but not the others (P1, P2). Wave parameters for these storm events, measured by deepwater and nearshore wave buoys, enable a comparison of storm characteristics and resulting beach–foredune impact. During the storm event which destroyed the foredune, nearshore wave height exceeded deepwater wave height, in contrast with other storms that year. The beach–foredune lost 78 m$^3$/m in 2022 and the notable 1974 storms that impacted this coastline resulted in 95 m$^3$/m volume loss. During 2023, beach recovery has occurred, but not rebuilt the foredune. It had persisted for ~40 years enduring many other severe storm events, and the coastal protection afforded by the dune system has been compromised. This highlights the need to consider dune morphology in assessments of erosion hazard and inundation risk along similar coastlines.

## Impact statement

This paper offers a fresh perspective on a long–term beach–foredune monitoring site in south-eastern Australia and presents the remarkable changes we observed in 2022. We present a robust dataset of beach–foredune monitoring accompanied by a unique combination of both deepwater and shallowwater wave observations which characterise a series of five storms that caused beach–foredune change. We note the differing impact of each of these storms and show how the most intense of these events caused wave overtopping of a foredune, while another event, around half as strong, actually removed this foredune. While subsequent recovery of sand to the beach has restored the shoreline to its previous position, the removal of the foredune means this section of coast is now more vulnerable to future wave impacts. The events of 2022 eroded 78 m$^3$/m of sand from the beach and foredune system and approaches the 95 m$^3$/m eroded in 1974 following the notable storms which impacted this region. In exploring the impacts on the beach and foredune and their causes, we shed light on the future of open sandy coastlines around the world and challenge readers to recalibrate their notion of expected coastal change.

## Introduction

There has been growing concern around the world for the future of sandy coastlines given that climate change will accelerate sea level rise (Dangendorf et al., 2019) and potentially increase the intensity and frequency of storm events (Reguero et al., 2019; Kaur et al., 2021). Global analyses have suggested the potential for widespread erosion and loss of beach and dune systems (Vousdoukas et al., 2020) with a rebuttal pointing to the dangers of overlooking regional and local-scale factors (Cooper et al., 2020; Short, 2022). Given this discussion, there is an urgent need to better constrain the dynamics of natural beach and dune systems to provide a critical baseline of understanding upon which to build future projections. Recent progress in extracting shoreline positions from satellite data has produced unparalleled regional and global timeseries of beach change (Nanson et al., 2021; Nanson et al., 2022; Vos et al., 2023a). Yet these 1D shoreline datasets contain horizontal uncertainties in shoreline position of ~10 m in microtidal settings, and greater uncertainties in meso- to macro-tidal beach environments (Vos et al., 2023b). They also do not capture the complexity of beach morphological and volumetric change in response to metocean conditions, nor do they consider the behaviour of dune systems which commonly back sandy beaches and

which interact with the beach. Thus, despite the utility of satellite-derived shorelines for regional assessments, decadal beach and dune morphodynamics, including storm erosion and recovery, must still be deduced from long-term topographic surveys or remote sensing techniques that retain 3D features of coastal landforms (e.g. photogrammetry) (Hanslow, 2007; Doyle et al., 2019).

Several multi-decadal beach–dune topographic survey programmes exist around the world in a variety of coastal settings, including the non-tidal southeastern Baltic coast at Lithuania (Jarmalavičius et al., 2012, 2017, 2020) and Poland (Różyński, 2005; Ostrowski et al., 2016), the Netherlands at Egmond aan Zee (Wijnberg and Terwindt, 1995; Rattan et al., 2005; Pape et al., 2010) and Noordwijk (Wijnberg and Terwindt, 1995; Kroon et al., 2008; Quartel et al., 2008), the US coast at Duck, NC (Larson and Kraus, 1994; Zhang and Larson, 2021), Rhode Island (Lacey and Peck, 1998), Torrey Pines (Ludka et al., 2019), the NW coast of the US (Ruggiero et al., 2016), Canada (Ollerhead et al., 2013), several beaches around the southwest of England (McCarroll et al., 2023), Porsmilin Beach (Bertin et al., 2022) and Vougot Beach (Suanez et al., 2023) in northwestern France, and the Hasaki coast of eastern Japan (Banno et al., 2020; Eichentopf et al., 2020). In southeastern Australia, two of the longest beach survey programmes in the world exist in micro-tidal, wave-dominated settings, one at Narrabeen-Collaroy from 1976–present (Turner et al., 2016), and another at Bengello Beach from 1972–present (McLean et al., 2023). Both these sites are repositories of multidecadal beach change with the Bengello site also capturing foredune dynamics and beach–foredune interaction over the survey period.

To accompany these two survey programmes, deepwater wave conditions along the southeast Australian coastline have been monitored for decades by the Manly Hydraulics Laboratory (MHL) using a network of wave buoys, with wave height and period records extending back to the 1970s and directional observations commencing progressively across the network from the 1990s. While the ocean wave buoy network measures deepwater wave conditions along the NSW coast, wave observations in shallowwater remain sparse and less accessible. To address that, a systematic programme of nearshore wave deployments in shallow coastal waters (<35 m) was commenced by the NSW Department of Climate Change, Energy, the Environment and Water (DCCEEW) in March 2016, which includes 20 observation locations to date (Kinsela et al., 2024). This data is being used to calibrate wave models to investigate and predict coastal hazards along the NSW coast. The longest deployments to date have been positioned adjacent to the long-term monitoring sites at Collaroy-Narrabeen Beach and Bengello Beach. The nearshore wave data enables new insights regarding wave transformation into the nearshore and its impact on beach and foredune change at these sites.

This study presents new data and observations of beach–foredune change at Bengello Beach in 2022. The foredune, which developed during the period covered by the 50-year survey programme, was severely eroded, overtopped and then destroyed due to the impact of storm wave conditions in 2022. Utilising the beach topographic data and photographic record, accompanied by deepwater and nearshore wave observations, this paper aims to explore the drivers of beach and foredune change during recent large storms and storm sequences and place these results within the context of multi-decadal trends in beach and foredune morphology and volume.

## Regional setting

Bengello Beach is a ~6-km-long sandy beach approximately 250 km south of Sydney on the NSW south coast (Figure 1). The shoreline is crescent shaped, faces ESE and is bounded in the north by the rocky Broulee Head with a tombolo connecting to Broulee Island. In the south, the beach is bounded by a training wall which directs the northern bank of the Moruya River estuary entrance. Bathymetric contours parallel the Bengello shoreline and the shoreface has a steeply concave geometry in the centre of the beach out to ~30 m water depth (Oliver et al., 2020). The beach is backed by a 2-km-wide strandplain comprising a series of ~60 foredune ridges formed over the mid- to late Holocene with radiocarbon and optically stimulated luminescence (OSL) dating studies constraining the shoreface and shoreline evolution, respectively (Oliver et al., 2015; Thom et al., 1981; Thom and Roy, 1985). OSL dating of foredune ridges comprising the outer ~150 m of the strandplain reveals continued progradation during the past ~500 years at a rate consistent with the Holocene trend of 0.27 m/yr. (Tamura et al., 2019).

McLean et al. (2023) have presented a comprehensive summary on the changes to Bengello Beach over 50 years (January 1972 to January 2022). The beach–foredune system at this site has been monitored monthly to bimonthly at four profiles located near the centre of the beach (Figure 1). These surveys documented the severe erosion events of the mid to late 1970s, the recovery from which built a new foredune 30–40 m seaward of the now degraded scarp (McLean and Shen, 2006; McLean et al., 2023). The beach has undergone cycles of erosion and recovery over the survey period, changing from more dissipative morphodynamic states to more reflective (Wright and Short, 1984). The beach surveys show that beach slope averages 4° (between MSL and +2 m) but fluctuates between ~2–7° depending on morphodynamic state and erosion and accretion due to storms. Since the early 1980s when the foredune developed, beach accretion and erosion cycles had occurred on the seaward side of this foredune. The foredune itself is vegetated and stabilised with pioneering species on the seaward side such as *Spinifex sericirus*, sea rocket *Cakile maritima* and *Cakile edentula* and coastal pigface *Carpobrotus glaucescens* dominating its crest and seaward side, while the landward side comprises secondary species such as coastal sword sedge *Lepidosperma gladiatum*, mat rush *Lomandera longifolia* and coastal wattle *Acacia sophorae*. The seaward side of the foredune has experienced numerous storm erosion events which generally create a scarp of 1–2 m. Post-storm recovery involves scarp slumping, backshore building from landward migration of sand due to aeolian transport and revegetation with the pioneering species. Aeolian sand transport most likely occurs under persistent ESE or ENE wind with velocities >28 km/h capable of transporting the average grain size found on the upper beach or berm (Doyle et al., 2024).

At Bengello Beach, prevailing waves are from the SSE to SE with an average significant wave height ($H_{sig}$) of 1.5 m and average peak wave periods are generally between 8 and 10 s. The intense storms, both tropical and extratropical, which produce large and powerful waves and low storm surges (by global standards), are the persistent cause of beach erosion along the eastern Australian coast. Storm waves in this region ($H_{sig} > 3$ m) are also typically from the SSE, and there were on average 15 storm events per year between 1986 and 2009 recorded by the Batemans Bay wave buoy. The average significant wave height for these storms was 3.71 m with an average maximum wave height of 7.19 m and an average duration of 57 h (Shand et al., 2010). Wave periods during storm events are typically between 10 and 15 s. Bengello Beach and the adjacent coastline experiences a mixed semi-diurnal micro-tidal regime with a spring and neap tidal range of 1.6 m and 0.7 m, respectively.

Metocean conditions in this region and hence beach–foredune erosion/recovery are known to be influenced by climate cycles,

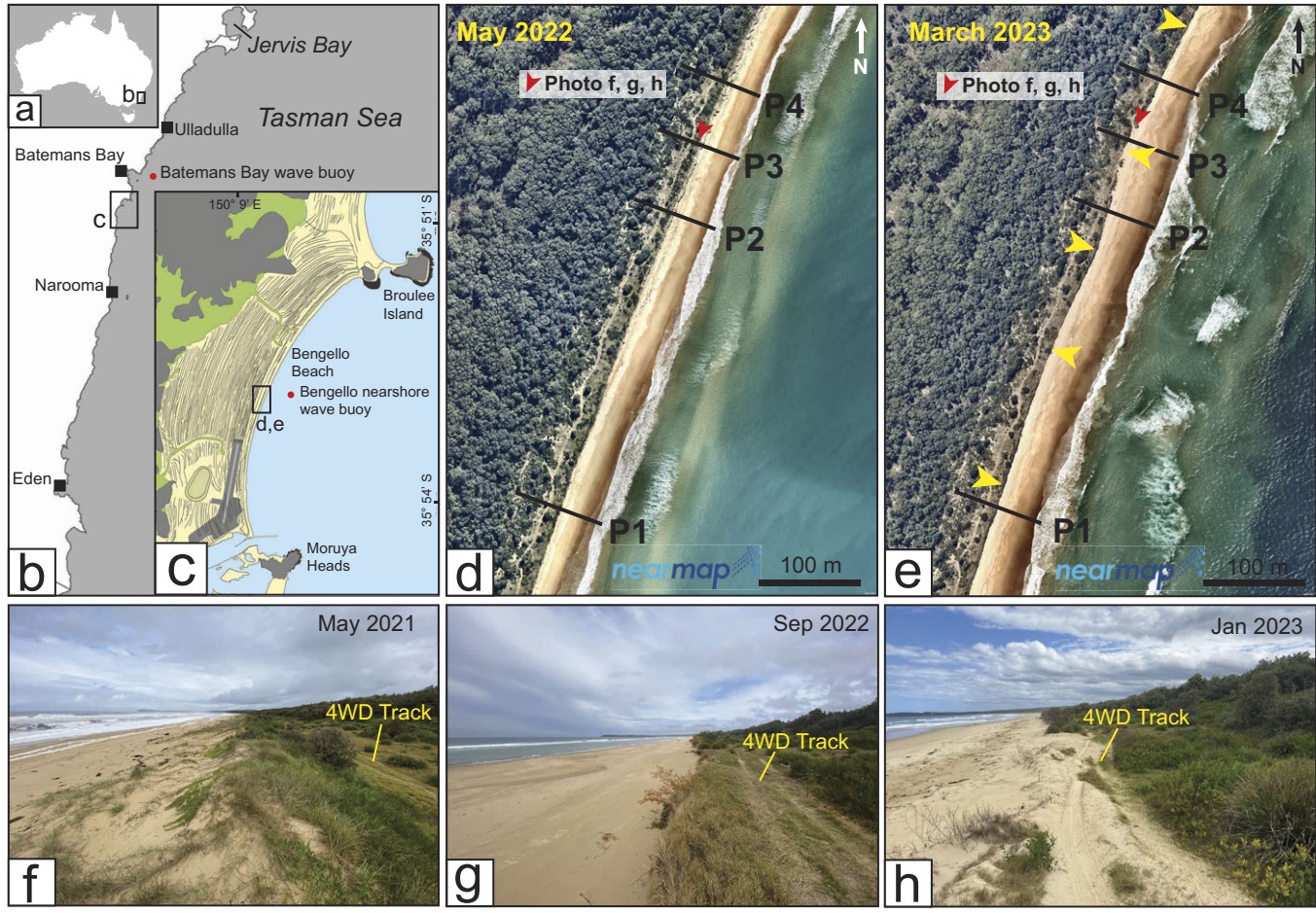

**Figure 1.** (a, b, c) Location of Bengello Beach in southeastern Australia and the location of the four profiles (P1–P4) monitored since January 1972 demarcated on Nearmap imagery from May 2022 (d) and March 2023 (e). Photos (f, g, h) showing the destruction of the foredune at Profile 3 (P3) with photo location and direction of view indicated in (d) and (e). Yellow arrows in (e) indicate the alongshore variation in foredune scarp position which developed in response to the July storms (Storms 4 and 5) and the associated megacusps which were present in the foreshore at this time (see Supplementary Figure 10).

especially the El Niño Southern Oscillation (ENSO) and the Southern Annular Mode (SAM) (Harley et al., 2010; Browning and Goodwin, 2013; Barnard et al., 2015; Mortlock and Goodwin, 2016). These are also known to influence one another (Gong et al., 2010; Lim et al., 2013). The Southern Oscillation Index (SOI) indicates the strength of the ENSO climatic pattern (Wang et al., 2017; Trenberth, 2020). When eastern Australia experiences a La Niña, there is generally increased rainfall and storminess, and during El Niño, rainfall and storminess is reduced. The Southern Annular Mode (SAM) also influences rainfall and storminess.

## Methods

### Survey methodology and beach–foredune metrics

Four profiles at Bengello Beach, which have been monitored monthly to bimonthly since January 1972, are labelled P1 to P4, with P1 separated from the other three profiles by 286 m and P2, P3 and P3 ~70 m apart (Figure 1d, e; McLean et al., 2023). Beach–foredune surveys in 2022 were conducted using an RTK GPS with each successive survey referencing to a series of datums. For this study, surveys are refenced to the Swale Datum (SD) and Foredune Datum (FD) at each profile with a Back Datum (BD) positioned further inland only relevant to the longer survey programme (see

McLean et al., 2023 for a fuller explanation of datums used at this site). Beach–foredune volumes were computed for each of the four profiles by taking the beach–foredune topography at the time of the survey and calculating area under the curve bounded by a horizontal line at 0 m Australian Height Datum (AHD) (which approximates mean sea level along this coastline) and a line extending vertically downward from the SD. Area under the curve ($m^2$) is converted to a volume ($m^3$) assuming a 1-m-wide profile. Beach–foredune volume over time was computed relative to January 2022. Change in the +3 m intercept relative to January 2022 was also calculated as the position of this contour broadly corresponds to the position of the beach–dune interface and is largely beyond the influence of fairweather wave processes. To place these results in the context of the longer-term survey programme presented in McLean et al. (2023), we added a fixed volume representing the profile further landwards of the SD to the BD where past change has occurred but is no longer part of the active beach–foredune zone.

### Deepwater and nearshore wave conditions

Wave buoys have been maintained immediately offshore of Batemans Bay in 65–84 m water depths continuously since May 1986, with the current position (−34.740278, 150.3175) in 65 m water depth (Figure 1b) occupied since February 2018. Non-directional

**Table 1.** Storm events of 2022 recorded by the Batemans Bay wave buoy and Bengello wave buoy. The cells highlighted by underlining show that for Storm 5, the Bengello nearshore wave buoy had a higher $H_{sig}$ than that of the Batemans Bay deepwater wave buoy, whereas for all other storm events in 2022 the Batemans Bay buoy $H_{sig}$ exceeded that of the Bengello buoy by ~1 m. Note that $H_{sig}$ here refers to the spectral significant wave height while $H_{max}$ is a time domain parameter calculated using zero upcrossing method. $T_p$ is the period associated with the frequency at the peak of the energy spectrum, that is, the frequency of highest energy density. For average direction, $D_p$ has been used which is the direction corresponding to the peak of wave energy (also a spectral parameter) and is the average value for the period during which $H_{sig}$ consecutively exceeds 3 m. Peak wave power is the peak value of the instantaneous wave power per metre alongshore which incorporates both $H_{sig}$ and $T_p$ to capture energy/power of the wave conditions. Cumulative storm wave energy flux for $H_{sig}$ > 3 m is a measure of the total wave power directed at the shoreline during the period when $H_{sig}$ exceeds 3 m and has been calculated following the method of Harley et al. (2017). Average wind strength and direction as well as rainfall is from the nearby Moruya Heads station. Peak TWL is shown for the March, April and July storm events (see Supplementary Figures 1, 2 and 3)

| Storm | Storm 1: 2–5 Mar | Storm 2: 8–10 Mar | Storm 3: 31 Mar – 4 Apr | Storm 4: 3–5 Jul | Storm 5: 10–11 Jul |
|---|---|---|---|---|---|
| **Deepwater waves** | | | | | |
| Peak $H_{sig}$ | 4.1 m | 5.0 m | 7.0 m | 4.3 m | <u>4.4 m</u> |
| Peak $H_{max}$ | 7.6 m | 9.7 m | 12.6 m | 7.8 m | 8.9 m |
| Peak $T_p$ | 13.8 s | 12.9 s | 14.9 s | 12.1 s | 16.0 s |
| Average direction $D_p$[1] | 87° E | 149° SSE | 148° SSE | 131° SE | 133° SE |
| Duration of consecutive $H_{sig}$ > 3 m | 48 h | 40 h | 64 h | 39 h | 31 h |
| **Nearshore waves** | | | | | |
| Peak $H_{sig}$ | 3.3 m | 3.5 m | 6.3 m | 3.4 m | <u>5.0 m</u> |
| Peak $H_{max}$ | 6.0 m | 6.4 m | 11.3 m | 5.5 m | 8.2 m |
| Peak $T_p$ | 12.8 s | 11.4 s | 14.6 s | 11.4 s | 14.6 s |
| Average direction $D_p$[1] | 91° E | 111° ESE | 114° ESE | 90° E | 114° ESE |
| Duration of consecutive $H_{sig}$ > 3 m | 4.5 h | 7 h | 43 h | 5 h | 26.5 h |
| Peak wave power | 142 kW/m | 141 kW/m | 586 kW/m | 128 kW/m | 372 kW/m |
| Cumulative storm wave energy flux for $H_{sig}$ > 3 m | 0.71 MW/Hm | 0.98 MW/Hm | 6.49 MW/Hm | 0.53 MW/Hm | 2.43 MW/Hm |
| **Atmospheric conditions** | | | | | |
| Predominant wind direction and strength | WSW ~30 km/h | WSW ~30 km/h | SW ~30 km/h | WSW ~30 km/h | NE ~60 km/h |
| Rainfall | 262 mm recorded from 1–10 Mar | | 18.4 mm | 46 mm recorded from 2–11 July | |
| **Total water level** | | | | | |
| P1 | | 2.4 m AHD | 4.0 m AHD | 3.4 m AHD | |
| P2 | | 2.9 m AHD | 4.4 m AHD | 4.0 m AHD | |
| P3 | | 2.8 m AHD | 3.9 m AHD | 3.8 m AHD | |
| P4 | | 2.8 m AHD | 3.8 m AHD | 3.9 m AHD | |

wave buoys were deployed at Batemans Bay until February 2001 when directional buoy (DWR-MkIII) deployments commenced (Kulmar et al., 2013). Deepwater wave data from Batemans Bay were obtained for the study period from MHL as hourly wave parameter time series including standard wave height, period and direction. A nearshore Sofar Spotter wave buoy has been maintained in 12–13 m water depth immediately adjacent to the Bengello Beach survey transects (−35.88000, 150.16108) since November 2020 (Figure 1c). The Spotter wave buoys use GNSS positioning and Doppler shift to measure their displacement on the water surface, and wave data are comparable to other standard wave buoy technologies (Kinsela et al., 2024). The data collection and processing methods have been described by Kinsela et al. (2024). The nearshore wave buoy data were analysed to compare the wave conditions (e.g. height, period, direction) between storm events observed at Bengello Beach in 2022 and to compare the offshore (deepwater) and nearshore wave conditions during each storm. Total water levels (TWLs) were also calculated at each profile

throughout the storm events. The M2 "model of models" formula of Atkinson et al. (2017) was used to calculate the 2% exceedance run-up level (Ru2%) including wave set-up. The beach slope used for each profile and event was the mean of beach slope values calculated between mean sea level (0 m AHD) and 2 and 3 m elevation using the pre- and post-event topographic surveys at each profile. Wave buoy data measured in ~13 m water depth adjacent to the profiles throughout the events were used to calculate Ru2% at each profile. The TWLs were then obtained using the calculated Ru2% values and ocean water levels measured at the nearby Batemans Bay ocean tide gauge.

## Results

### Storm events at Bengello Beach in 2022

Five storm events resulting in significant beach–foredune change were observed at Bengello Beach during 2022 and are analysed here.

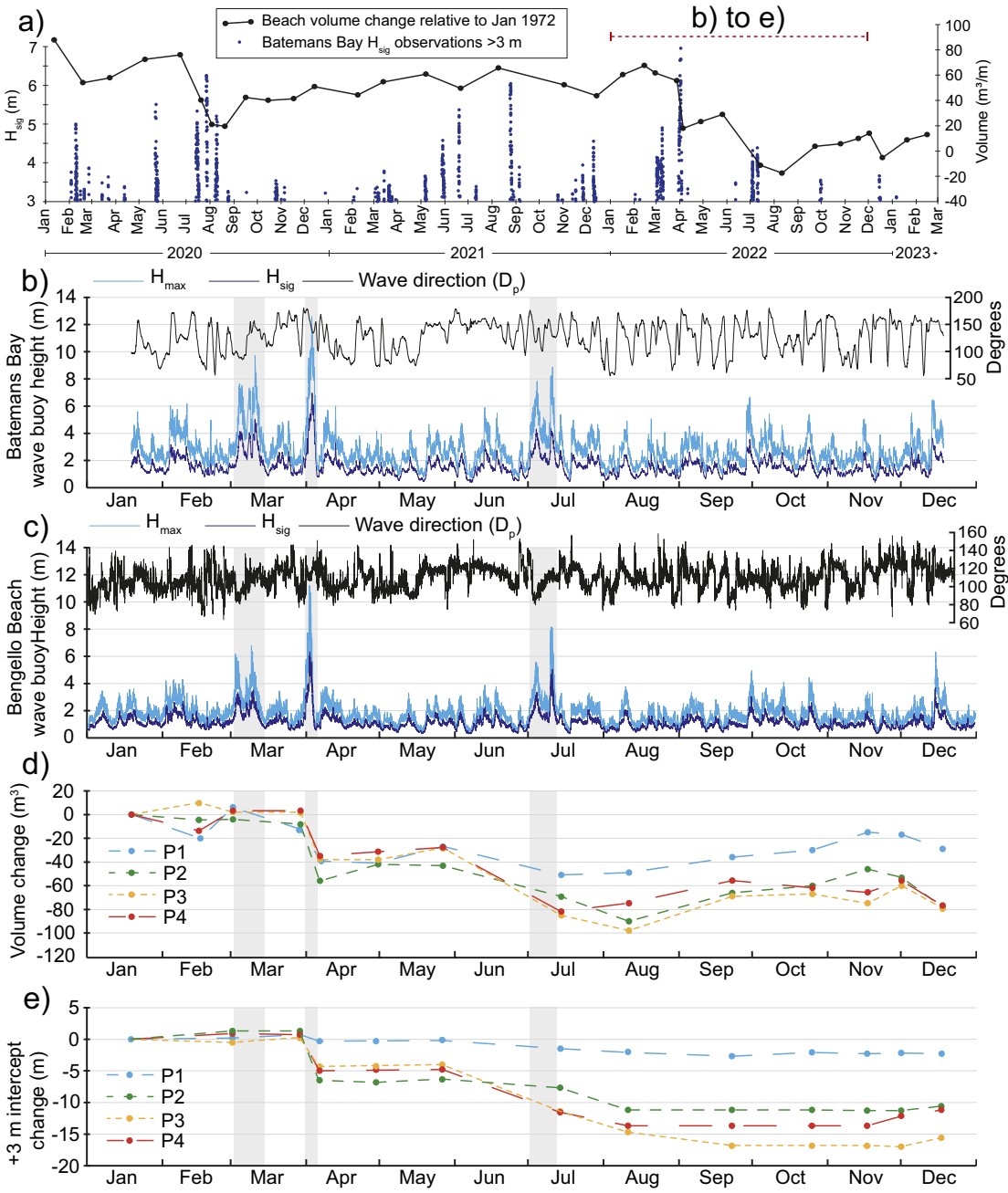

**Figure 2.** (a) Hourly $H_{sig}$ observations >3 m recorded by the Batemans Bay wave buoy for the period January 2020 through to March 2023 accompanied by beach volume change over the same time period relative to January 1972. (b) Recorded deepwater ocean wave conditions from the Batemans Bay wave buoy for the 2022 including significant wave height ($H_{sig}$), maximum wave height ($H_{max}$) and wave direction (degrees). (c) Recorded nearshore wave conditions from ~13 m water depth adjacent to Bengello Beach including significant wave height ($H_{sig}$), maximum wave height ($H_{max}$) and wave direction (degrees). (d) Change in beach volume over 2022 relative to the volume of the January survey for the four central beach profiles at Bengello. (e) Change in distance from the back datum to the +3 m intercept for each of the four central beach profiles at Bengello relative to the position in January 2022.

The first of the storm events of note were consecutive moderate storms (Storms 1 and 2; Table 1) which occurred in early March with peaks on 3 March and 9 March (Table 1; Figure 2b, c; Supplementary Figures 1 and 8). The second of these two events was slightly larger, and peak wave energy was from a slightly more southerly direction (Table 1). Also, during early March over the period corresponding to Storms 1 and 2, a moderate flood event in the nearby Moruya River brought with it both driftwood and a fine brown silt that covered the backshore of the beach (262 mm of rain

recorded during this period; Table 1). Peak TWL during these events were lowest at P1 (2.4 m AHD) and highest at P2 (2.9 m AHD) (Table 1).

Only weeks later, a more intense event (Storm 3) occurred between 31 March and 4 April 2022 which had the highest peak and total wave power of the five storms (Table 1; Figure 2b, c; Supplementary Figures 2 and 8). During this event the Batemans Bay buoy recorded a $H_{sig}$ > 6 m for ~8 h which coincided with a spring high tide (Supplementary Figure 2). The $H_{max}$ on the

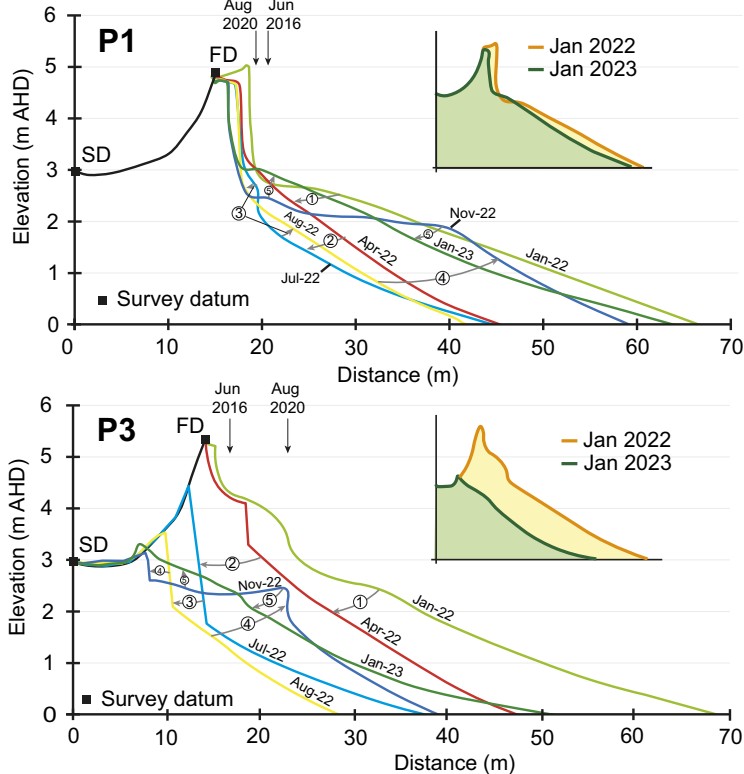

**Figure 3.** Selected beach surveys from Profile 1 and Profile 3 showing changes to the beach morphology between January 2022 and January 2023. Vertical black arrows indicate the position of scarps which developed as a result of the June 2016 storm events and the storms in August 2020.

Batemans Bay buoy also peaked above 10 m with a value of 12.6 m closely corresponding with this high tide. Wave direction at the onset of the storm was between 170° and 180°, but at the time of peak, wave heights were southeasterly between 130° and 140°. During this event, the Bengello nearshore wave buoy measured $H_{sig}$ values of 4.5–6 m. During the high tide, $H_{max}$ exceeded 8 m. Wave direction recorded by the Bengello nearshore wave buoy was aligned with the orientation of the beach (average of 114°), and there was no notable shift in direction during the event. Peak TWL was between 3.8 and 4.4 m across all four profiles during Storm 3 (Table 1).

The third group of storms of note occurred between the 1 and 13 July 2022 (Storms 4 and 5; Table 1; Figure 2b, c; Supplementary Figure 3). During the second event (Storm 5), the peak $H_{sig}$ at Bengello Beach actually exceeded the deepwater $H_{sig}$ value recorded by the Batemans Bay buoy (Table 1). This contrasts with the other events in 2022 where wave heights were generally ~1 m lower at the Bengello buoy compared to the Batemans Bay buoy (Table 1). Local storm generated wind sea from a prevailing onshore wind may be responsible for this higher value (Supplementary Figure 8). During this storm, nearshore wave direction was closely aligned with the orientation of the beach (Supplementary Figure 3). Also, when wave steepness was considered, this storm stood out from the others. Peak TWL for these events (Storms 4 and 5) was lowest at P1 (3.4 m) and higher at the other three profiles (3.8–4.0 m).

### Beach and foredune morphological changes in 2022

In January 2022, a degraded scarp was evident at all four profiles – a legacy of storm events in 2020. In the case of P1, the scarp in 2020

was <1.5 m landward of the scarp which developed as a result of the June 2016 storm (Figure 3), the most significant regional beach erosion event of the past decade (Harley et al., 2017). In contrast, at the other three profiles, the degraded scarp from 2020 was 4–6 m further seaward than the June 2016 scarp (Figure 3) due to beach recovery. Beach profiles in January 2022 had a gently concave profile with a subtle berm appearing in the February surveys (14 and 28 February; Supplementary Figure 4). Importantly, while the storm events between 1 and 14 March (Storms 1 and 2) did not result in a substantial reduction in beach volume (Figure 2d), the beach profile was modified to a concave geometry (Supplementary Figure 4). This concave profile featured a ramp-like morphology that could be more conducive to wave runup amplification, potentially promoting foredune overtopping (Holman and Guza, 1984).

During the intense storm event of early April (Storm 3; Table 1), wave overtopping of the foredune occurred and the driftwood on the back of the beach, brought by the March floods during Storms 1 and 2, was carried over the 5-m-high frontal dune and into the swale behind (Supplementary Figure 5). Interestingly, at P1 there was minimal overtopping, and debris was instead deposited at the base of the scarp, which was almost 2 m high and present in all surveys prior to the event (17 January to 28 March). Immediately prior to Storm 3 in early April and in contrast to P1, the other profiles (P2, P3 and P4) were all in a healthy condition with low mounds of sand covered by *Spinifex sericirus* extending several metres seawards of the degraded and vegetated scarp from events in 2020. Three things are significant here; first, there was no sand carried over the foredune with the driftwood; second, there was no evidence of any backwash or return flow; and third, damage to the beach was only moderate, with the Spinifex-covered backshore trimmed back by ~5 m at P2, P3 and P4 creating a ~ 0.5–1.5-

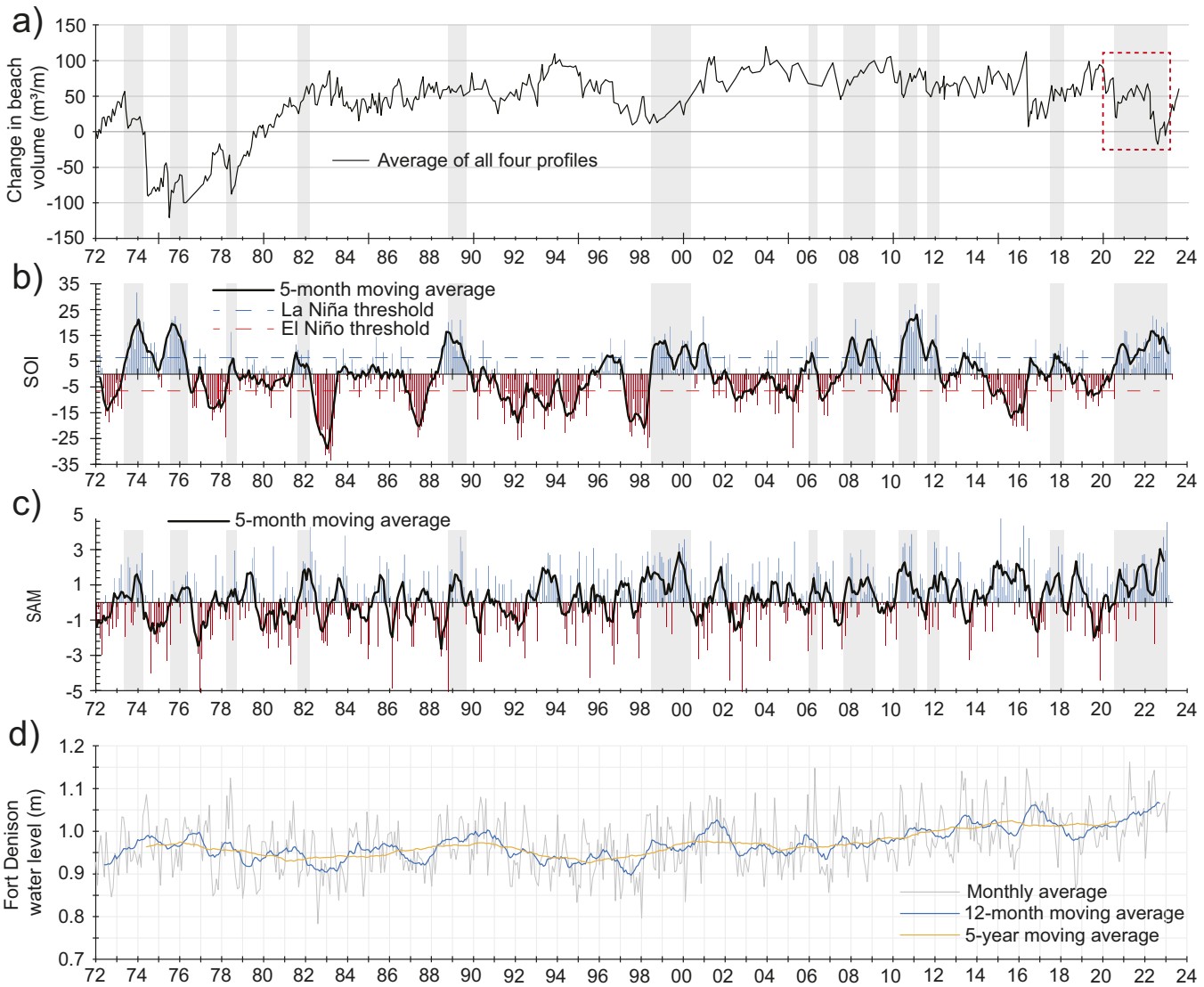

**Figure 4.** (a) Ensemble beach volumes relative to starting volume in January 1972 through to August 2023 from McLean et al. (2023). (b) Southern Oscillation Index (SOI) from 1972 to 2023 where sustained positive SOI values, especially above the threshold, indicate La Niña conditions while negative SOI values below the threshold indicate El Niño conditions. The data has been fitted with a 5-month moving mean. (c) Southern Annular Mode (SAM) index from 1972 to 2023 after Marshall (2003) where during positive phases of SAM, the strong westerly winds of the mid to high southern latitudes shift south which generally increases the rainfall in southeastern Australia. During negative phases of SAM, this belt of strong westerly winds shifts northward decreasing rainfall in southeastern Australia. There are differences in the distribution of rainfall during the positive and negative phases of SAM depending on where the positive or negative occurs in summer or winter. (d) Monthly average water levels from the Fort Denison tide gauge in Sydney with a 12-month moving average and a 5-year moving average plotted through the data.

m-high scarp while at P1, the existing ~2-m-high scarp shifted subtly inland by ~1 m. At all profiles, the beach was planed down and steepened as a result of this event. Following that event until the end of June, the beach recovered slightly, building vertically and seaward under modal wave conditions (Figures 2b, c and 3).

The beach–foredune survey of 14 July 2022, immediately after the storm events that occurred in early July (see Section 3.1 above), revealed the loss of the foredune datums at P3 and P4 as the +3 m intercept shifted ~7 m inland, removing a large portion of the foredune complex (Figure 2e). In contrast, there was no appreciable change in the position of the pre-existing scarp at P1 (Figure 3), also reflected in the stability of the +3 m intercept (Figure 2e). During the July storms the foredune scarp and foreshore at the profile locations developed a distinctive crenulate morphology resembling megacusps with indentations spaced 250–300 m apart which persisted into August 2022. While the beach morphology has since lost

this crenulate morphology, it is still visible in the position of the foredune toe/vegetation line even in 2023 (Figure 1e; Supplementary Figures 10 and 11). The reasons for this consistent but relatively small-scale variability is discussed below.

During the remainder of July and throughout August and September, there were no further storm wave events and yet beach–foredune surveys in August show further landward shifts in the position of the +3 m intercept at P2, P3 and P4 (Figure 2e). The November survey recorded a defined berm between +2 and 2.5 m at all four profiles indicating the beginning of beach recovery (Figure 3; Supplementary Figure 6). Sand from this berm was starting to move into the backshore and the process of rebuilding has continued throughout 2023 with a berm achieving its maximum dimensions of 15–20 m width and ~ 2.1–2.2 m height in November 2023. In December 2023 and January 2024, this berm has been again planed down but substantial transfer has occurred

landwards to recover the backshore and repair the scarp left by the 2022 events. Despite this rebuilding phase, the dune morphology of P2, P3 and P4 is very different with the foredune partially removed at P2 and completely removed at P3 and P4 (Figure 3; Supplementary Figures 6 and 9).

### Beach volume change in 2022

The ramp morphology produced after the March events (Storms 1 and 2) and the overtopping, backshore trimming and beach steepening caused by the early April event (Storm 3) had the least impact in terms of volume on P3 and P4. Overall, the volume change from Storms 1 and 2 was minimal while the impact of the event in early April 2022 (Storm 3) eroded an average of 38 m$^3$/m from the beach–foredune (average volume loss from P1–P4). The beach stabilised and recovered slightly, before the back-to-back storms in early July (Storms 4 and 5) caused more beach–foredune erosion, such that by August, on average, a further 47 m$^3$/m of sand had been removed. Thus by mid-August 2022, a net volume of approximately 78 m$^3$/m of sand had been eroded from the beach–foredune system.

What is especially striking about the changes observed at Bengello Beach in 2022 is the different behaviour of P1 compared with P2, P3 and P4. While P1 lost some sand, especially after Storm 3, subsequent volume change was relatively modest compared to P2 and especially P3 and P4. Comparing the volume change observed for P1–P4 between the January 2022 survey and the survey in mid-August 2022, we see that P1 lost 49 m$^3$/m, P2 lost 90 m$^3$/m, P3 lost 98 m$^3$/m, P4 lost 75 m$^3$/m.

## Discussion

### Temporal and spatial variability of storm impacts

The beach–foredune sand loss that occurred in 2022 appears to be a culminating phase of erosion events which began in 2020. Storms in February and July–August of 2020 removed ~70 m$^3$/m of sand from the beach–foredune (Figure 2a). The recovery phase during late 2020, through 2021 and into the beginning of 2022, was only modest, such that beach–foredune volume in early 2022 (January–February) had not returned to the 2020 level (Figure 2a). Thus, the events of 2022 in the context of the previous 2 years (2020–2021) meant that the impact of the storms in April and July 2022 achieved what significant storms in previous years had not – the destruction of the foredune at two of the four profiles and the lowest beach–foredune volumes observed since June 1979 (Figure 4a; McLean et al., 2023). Since the early 1980s when the foredune developed, all change occurred on the seaward side of this foredune, and now for the first time since, there is wave influence reaching the swale formally sheltered by the foredune.

The five storms in 2022 had differing impacts on the beach–foredune system. Storm 3 had the greatest wave power and wave direction, was aligned with the shoreline and also the highest TWL (Table 1). This resulted in foredune overtopping at all profiles, although only 38 m$^3$/m of erosion on average. Storm 5 caused the greatest morphologic impact to the beach–foredune, and this storm stands out from the others, as although it had moderate wave energy, nearshore wave heights exceeded deepwater wave heights and it was the only storm that had strong and persistent onshore winds (Table 1; Supplementary Figure 7). It is worth considering the duration of the five storms, as both Storms 3 and 5 stand out

considering cumulative storm wave energy flux for H$_{sig}$ > 3 m (Table 1), although Storm 5 which caused foredune destruction is still only half as powerful as Storm 3 using this metric. Variability in TWL between the four profiles during the July storms (Storms 4 and 5) may have contributed to differing beach–foredune impact and erosion volumes by controlling the intensity of wave attack of the dunes. During these events in July, P1 had the lowest TWLs (0.3–0.5 m lower than the other profiles; Table 1) and experienced minimal foredune erosion, while P3 experienced the most (see Section 4.2 above). In contrast, during Storm 3 (April), the TWLs calculated at the four profiles were reasonably consistent and foredune overtopping and moderate erosion occurred at all profiles.

Thus overall, although Storm 3 (April) was more powerful and had higher TWLs than the others (Table 1), the July storms produced more dramatic morphological changes to most of the profiles (Figure 3). Others have noted how a relative lower-energy storm event may result in substantial beach–foredune erosion due to the synchronisation of waves, tides and winds (Guisado-Pintado and Jackson, 2018). Furthermore, Rangel-Buitrago and Anfuso (2011) show that more moderate storm events can still produce important morphological changes to the berm and foreshore while more severe events impact the foredune. In 2022 at Bengello, Storms 1 and 2 removed a berm and lowered the foreshore, enabling foredune erosion and overtopping in Storm 3, and foredune destruction at several profiles in Storms 4–5. Thus morphological "work" was achieved even with moderate storm events and likely enhanced the impact of later more severe events emphasising the importance of antecedent beach conditions (Splinter et al., 2014).

The spatial variability of the impact of Storm 5, expressed in the crenulate scarp and beach–foredune megacusps, may have been influenced by variation in dune vegetation (species, condition, percent coverage) and overall dune morphology (Davidson et al., 2020), although in this instance the rhythmicity of the crenulate scarp and its expression in the foreshore suggests beach and surf zone morphodynamics are more likely. Castelle et al. (2015) note the importance of megacusps in controlling variable dune erosion whereby erosion is exacerbated at the head of the megacusp embayment and state that antecedent morphology of the surf zone bars is important. Megacusp development leading to variable profile response to Storms 4 and 5 at Bengello may have resulted from the development of rip embayments just prior to these events as shown by Sentienel-2 satellite images. These images also show a rip embayment that persisted throughout July and August adjacent to P3 and led to further landward migration of the foredune scarp (Figure 3; Supplementary Figures 6 and 10).

### Climatic conditions 2020–2022

It is worth considering how the climatic conditions corresponding to the 2020–2022 period may have contributed to the observed changes at Bengello Beach. Although the foredune has been regularly scarped by storm events since its development, the survey programme has not documented such drastic change as the destruction of the foredune itself. Figure 2 shows two timeseries of relevant climatic indices which influence metocean conditions in this region (Harley et al., 2010; Browning and Goodwin, 2013; Barnard et al., 2015; Mortlock and Goodwin, 2016). These climatic patterns have been linked to shoreline behaviour over both local (Ibaceta et al., 2023) and regional spatial scales (Vos et al., 2023a). Considering the three-year period from the beginning of 2020 to the end of 2022, a strong la Niña phase (positive SOI) is indicated and was popularly described as a "triple dip" La Niña. An

accompanying "triple dip" positive SAM whose peaks corresponded to the austral spring–summer seasons also occurred during this period. The combined period of overlap was from October 2020 through to March 2023 totalling 30 months (SOI and SAM 5-month moving averages >0). This combined positive SOI (La Niña phase) and positive peaks of SAM have occurred at other times, although in many cases these two indices are out of phase. Where they are aligned, beach–foredune response is variable. The two other periods where they corresponded for the longest time actually show accretion (Figure 4). However, several shorter periods of overlap do correspond to erosion, for example, during the 1970s (Figure 4a–c).

Recent studies have suggested links between these climate cycles and more energetic wave conditions for southeastern Australia. For example, Marshall et al. (2018) show that positive phases of SAM during austral summer appear to produce a slight increase in $H_{sig}$ along the southeastern coast of Australia as more wave energy propagates into the Tasman Sea. Godoi and Torres Júnior (2020) show that when positive SAM in austral summer corresponds with a La Niña phase, there is an increase in $H_{sig}$ of between 0.2 and 0.4 m in the northern Tasman Sea and increase of between 0.3 and 0.6 s in wave period along the length of the NSW coast. Studies have also associated changes in the frequency of extreme events in this region with changing climatic conditions. For instance, Browning and Goodwin (2013) have shown that extratropical cyclones which form and intensify in the Tasman Sea, and are associated with severe beach erosion along this coastline, occur more frequently during positive ENSO. Overall, the correlation between what could be termed the "double triple dip" (three consecutive positive SAM phases during summer combined with three consecutive phases of La Nina) and the response of Bengello Beach is at present a correlation, not causation. However, it is certainly an intriguing one.

### The future for Bengello Beach

For Bengello Beach and other shorelines of this region, we note the threats posed by projected climate change influencing wave height and direction with potential for intensification of seasonal and climatic patterns (Liu et al., 2023) as well as the impact of projected sea level rise over the coming century. The Fort Denison tide gauge recorded a sea level rise of 2.5 mm/yr over the past ~20 years (Figure 4d), and McLean et al. (2023) noted a subtle but steady decline in beach–foredune volume from ~2010 onwards. The events of 2022 have further extended this trend (Figure 4a).

Arriving at Bengello Beach in 2022 soon after the July storm events, we were surprised to find the foredune removed at two profiles. (We use the term "surprise" deliberately, defined as a "low-likelihood" event (Chen et al., 2021, p. 203).) We anticipated that the foredune, which developed in the 1980s, would persist into the future. The broader historical and geological context supported this view. Firstly, the contemporary foredune had persisted for the past 40 years despite many other severe storm events and was a well-established feature of the profile morphology. Secondly, at this site, foredunes have been shown to persist for >100 years before being stranded behind another (Oliver et al., 2015). While it is possible that destruction and rebuilding could occur during the ~100-year foredune evolution, it has not been evident from detailed morphostratigraphic studies (Oliver, 2016; Tamura et al., 2019). Thus, what happened in 2022 at Bengello Beach was a surprising morphologic outcome and an abrupt change in the beach–foredune morphology. At a site where foredune building has been documented over millennia, centuries and decades (Oliver et al., 2015; Tamura

et al., 2019; McLean et al., 2023), foredune *destruction* is a profound outcome and raises the question: are we seeing the beginning of a system state tipping point being reached? If this is the case, there may be a need to recalibrate expectations on the future of sandy shorelines.

### Conclusion

This study has documented the dramatic change in beach–foredune morphology at Bengello Beach during 2022. The results show that a series of five storms from March to July caused foredune overtopping and beach erosion culminating in the removal of the foredune at two of the four profiles. Deepwater and nearshore wave recordings from these five events show that differences in wave power, duration and direction were related to beach–foredune response with overtopping and erosion occurring in April and foredune destruction occurring in July. We also found that Profile 1, which is only ~350 m south of Profile 3, behaved very differently in response to the same wave forcing. Overall, the events of 2022 appear to be a culminating phase of beach–foredune response to the period from 2020 to the end of 2021, where insufficient recovery occurred between successive storm events, exposing the foredune toe to repeated wave impact. Broader climatic conditions may have promoted more energetic wave conditions in the Tasman Sea leading to these successive storms and lack of time for beach recovery. This means that, looking to the future, modelling of storm demand for beaches needs to be nuanced to such a degree as to incorporate this variability. Assessments of foredune morphology are also critical in understanding erosion risk. Furthermore, there is a need to better understand beach recovery including its rates and style, so more tailored adaptation measures can be developed for a changing climate.

**Open peer review.** To view the open peer review materials for this article, please visit http://doi.org/10.1017/cft.2024.8.

**Supplementary material.** The supplementary material for this article can be found at http://doi.org/10.1017/cft.2024.8.

**Data availability statement.** The beach profiling data that support the findings of this study are available from the corresponding author, TO, upon reasonable request. Supplementary data has been provided to further support the main article. SAM monthly index data is available from https://legacy.bas.ac.uk/met/gjma/sam.html. SOI data is available from the Commonwealth of Australia Bureau of Meteorology via www.bom.gov.au/climate/enso/soi/. Water level data from Fort Denison tide gauge is available from www.bom.gov.au/oceanography/projects/ntc/monthly.

Deepwater wave data from the Batemans Bay Waverider buoy were collected and provided by Manly Hydraulics Laboratory on behalf of the NSW Department of Climate Change, Energy, the Environment and Water (DCCEEW) through the NSW Coastal Data Network Programme. Data are available on request to MHL. Nearshore wave data from the Bengello wave buoy were collected and provided by the NSW Department of Planning and Environment's Coastal and Marine Science Team. Nearshore wave data are available from the NSW Sharing and Enabling Environmental Data (SEED) portal: https://datasets.seed.nsw.gov.au/dataset/nsw-nearshore-wave-buoy-parameter-time-series-data-completed-deployments.

**Acknowledgements.** We wish to thank Brad Morris (Coast and Marine team, DCCEEW) for processing and QA/QC of raw shallowwater wave buoy data. We are grateful to Eurobodalla Shire Council for continuing to allow access to the survey site. We thank Prof. Sarah Perkins-Kirkpatrick for advice and direction regarding relevant literature on climatic indices in this region.

**Author contribution.** All authors have made substantial contributions to this submission. Dr. Oliver and Prof. McLean conducted the land-based fieldwork

and analysis and Dr. Oliver drafted much of the paper and created the figures. Dr. Kinsela and Dr. Doyle led the on-water fieldwork and analysed the wave and wind data and helped draft the methodology and results sections dealing with this data as well as substantially revising the discussion. Drs. Oliver, Kinsela and Doyle created the supplementary figures which support the paper.

**Financial support.** This research received no specific grant from any funding agency, commercial or not-for-profit sectors. The School of Science at UNSW Canberra supported the field components of this research.

**Competing interest.** The authors declare no competing interests exist.

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
