## [Editor Report]

Dear authors,

I have read your paper and the reviews of the other 2 reviewers. I agree with the sentiments of the reviewers and think more could be done to tighten up this story you are presenting here. It is a fascinating data set. Why focus just on 2022 (as asked by R2) or if you only focus on this, why include such a discussion on the longer term data set (R1). Flipping back an forth can blur the story that you are trying to tell your readers. Focussing on the 2022 storms, it would be great to see more information to correspond to the inshore wave forcing/dynamics to help put the variability a bit more into context. Similarly, given we are now a year out (still not much), recovery could use a bit more discussion I feel. 

Overall I think the paper can provide a nice contribution to the literature and upon addressing the comments of myself and your reviewers could be considered for this journal. 

All the best,

Kristen Splinter

Senior Editor, Coastal Futures

---

## [Editor Report]

Handling Editor Comments:

Foredune erosion, overtopping and destruction in 2 2022 at Bengello Beach, southeastern Australia

Dear Dr. Oliver and co-authors,

Thank you for your resubmission. As noted by the 2 reviewers, you have done a reasonable job at addressing the comments provided by both. One is happy with the manuscript as is and the other still requests some reasonable changes. I note that in several of your responses you cite the word limitation as an inhibitor for being able to address the requested change. This is something I will bring back to the management as I feel that paper could still be improved if more of these suggestions could have been included. Overall I am recommending this is returned for additional Minor revisions.

A key point made in your response was the focus on the 2022 event and that reframing of the paper, with adequate references to the McLean (2023) work. In taking this into account, I would suggest lines 21-23 (“The development of this same foredune has been documented by the survey program and thus we lost a foredune we watched being built. This survey program is thus to our knowledge an unprecedented record of both foredune development and destruction.”) should be removed as they refer to the longer survey program that is not the focus of this work. Rather, if backed up by the data, perhaps you could say that the foredune section of the beach was at its lowest volume with the 40+ year record of data available at this site. I would welcome a 1-line comparison to the 1974 storms that are historically known in this area for having been quite destructive. I have to loosely assume without looking at the data that 1974 may have been the last time the foredune was completely lost? This is somewhat alluded to in the abstract but I think you could make direct reference to the 1974 series of storms here as they were quite notable. This would indeed be a great add to the impact statement. However I note that in your discussion (5.1) you say that it’s the late 1970s when it was last this small wrt foredune volumes so perhaps nothing can be said about 1974. 

I would also suggest the tone could be softened a little in the abstract. The word drastically, exceptional, etc are a bit sensationalist. eg. L36-38: “Beach recovery has since occurred, but not yet rebuilt the foredune and the overall morphology at two of the four profiles is drastically different. The different profile response was exceptional with respect to the >50-yearmonitoring program which had also documented the building of this foredune”. Too much of this and it starts to read more like a sensationalist news article rather than an academic journal article in my opinion. 

As noted by R2 – subheadings should include more than a paragraph (eg. 5.1). Consider less headings and how this can be framed into more general themes. 

L327 – please don’t use authors initials here to refer to yourselves. I would just say the co-authors (or we) Also L413.

5.2 – I would consider how the forcing can be included within your results to present these together. As noted by the reviewers, estimates of TWL (tides+non-tidal residuals+setup+R2) are key to understanding the dune/foredune erosion and I still feel should be included. This helps to provide the ‘why’ it has happened rather than just presenting the beach volume changes. Max TWL will also change from profile to profile, also providing some assistance. Your co-author (Kinsela) should also have access to the higher resolution NSW wave model to get estimates of waves at each profile and not just at a buoy location. The lack of this analysis is still a major weakness of this work in my opinion. 

5.3 – I appreciate you say you can’t add more, but if your data shows that there was no time/ability for the beach to recover between these events and I think there is a case for clustering to be a driver here. 

5.4 – even if bathy is not available, satellites (Nearmap, Planet, or Landsat) could all help provide some insight here into pre-storm conditions. 

L413: why is “surprised” in italics? Are scientists allowed to be surprised? These were big events. Some level of impact had to have been expected.